# Cell Type-Specific TGF-β Mediated EMT in 3D and 2D Models and Its Reversal by TGF-β Receptor Kinase Inhibitor in Ovarian Cancer Cell Lines

**DOI:** 10.3390/ijms20143568

**Published:** 2019-07-22

**Authors:** Wafa Al Ameri, Ikhlak Ahmed, Fatima M. Al-Dasim, Yasmin Ali Mohamoud, Iman K. Al-Azwani, Joel A. Malek, Thasni Karedath

**Affiliations:** 1Department of Genetic Medicine, Weill Cornell Medicine-Qatar, Education City, Qatar Foundation, Doha P.O. Box No. 24144, Qatar; 2Sidra Medicine, Doha P.O. Box No. 26999, Qatar; 3Genomics Core, Weill Cornell Medicine-Qatar, Education City, Qatar Foundation, Doha, Qatar

**Keywords:** TGF-β, ovarian cancer, EMT, SKOV3, 3D models

## Abstract

Transcriptome profiling of 3D models compared to 2D models in various cancer cell lines shows differential expression of TGF-β-mediated and cell adhesion pathways. Presence of TGF-β in these cell lines shows an increased invasion potential which is specific to cell type. In the present study, we identified exogenous addition of TGF-β can induce Epithelial to Mesenchymal Transition (EMT) in a few cancer cell lines. RNA sequencing and real time PCR were carried out in different ovarian cancer cell lines to identify molecular profiling and metabolic profiling. Since EMT induction by TGF-β is cell-type specific, we decided to select two promising ovarian cancer cell lines as model systems to study EMT. TGF-β modulation in EMT and cancer invasion were successfully depicted in both 2D and 3D models of SKOV3 and CAOV3 cell lines. Functional evaluation in 3D and 2D models demonstrates that the addition of the exogenous TGF-β can induce EMT and invasion in cancer cells by turning them into aggressive phenotypes. TGF-β receptor kinase I inhibitor (LY364947) can revert the TGF-β effect in these cells. In a nutshell, TGF-β can induce EMT and migration, increase aggressiveness, increase cell survival, alter cell characteristics, remodel the Extracellular Matrix (ECM) and increase cell metabolism favorable for tumor invasion and metastasis. We concluded that transcriptomic and phenotypic effect of TGF-β and its inhibitor is cell-type specific and not cancer specific.

## 1. Introduction

Ovarian cancer is one of the most lethal gynecological malignancies, which accounts for 5% cancer deaths among women [1]. The overall survival of ovarian cancer patients is less than 30% as its diagnosis occurs after the metastatic spread with a very high rate of recurrence and chemoresistance [2,3]. The long term cure for the later stage cancer is challenging as little is known about the underlying mechanisms promoting ovarian cancer progression [4]. Most of the cancer studies relay on in vitro tumor models, like established cell lines and mouse xenograft models. Most of the tumor models fail to explain the aggressive phenotype represented in the real situation. These models usually help to identify the tumorigenicity of cancer cells and the involvement of the surrounding microenvironment. However, these systems are usually not sufficient to explain the initial metastatic event cascades, like tumor invasion and EMT [5]. Studying these events is crucial as it may lead to a better understanding of key metastatic events occurring primarily. Also, it may help to identify important molecular targets involved in increasing metastatic potential of some tumor subtypes. Identification of a sensitive molecular target controlling invasion and metastasis events that lead to EMT remains elusive. The lack of proper model system to study EMT and invasion is another limitation in this research area.

The in vitro cancer cell line models of 2D cultures are not an ideal system to explain the molecular mechanisms controlling EMT, metastasis and chemoresistance. Moreover, 2D models usually lack biological requirements which explain hypoxia and angiogenesis. When cancer cells grown in 2D, they have a uniform and rapid proliferation across the plastic surface unlike the 3D models that show many zones of differential proliferation and cell division, mostly confined to outer zones [6,7]. The growing cells in 3D capture more phenotypic heterogeneity due to difference in growth, access of nutrients and oxygen leading to selection pressure and resembling in vivo tumor organoids. Since tumor spheroids have oxygen and nutrient grades, 3D models could be established to test these theories with greater ease than serially passaged 2D models. The significance of 3D spheroid model systems has been comprehensively reviewed and clearly illustrated that cell-matrix interactions are better recreated by a complex aggregated cell population rather than a simple cell monolayer 2D culture [8,9,10,11]. It has been reported that in vitro 3D model of epithelial ovarian carcinoma (EOC) restores the functional differentiation of the tissue in vivo in a manner that cannot be achieved in the 2D monolayer culture [8,9,12]. The organotypic 3D models considerably reduce time and cost of drug discovery too [13]. The current study focuses on 3D models of ovarian cancer cell lines and their response towards prolonged exposure to the transforming growth factor beta (TGF-β).

It is a well-known phenomenon that TGF-β plays an important role in the tumor progression and actin cytoskeleton reorganization [14]. The role of TGF-β in inducing cell proliferation, invasion and migration in 3D culture is rarely studied in ovarian cancer. The present study evaluates the role of TGF-β in cell growth, invasion and metastasis and whether it may induce a complete EMT in both 2D and 3D cultures irrespective of the proliferation rate. The more invasive phenotypes developed by continuous TGF-β treatment and its role in cell adhesion are as well deciphered in the current study. The inhibitory effect of the selective inhibitor TGF-β RI (LY364947) in ovarian cancer cells has been analyzed. The present study evaluates the role of LY364947 in controlling cell proliferation and invasion of cancer cells.

## 2. Results

### 2.1. TGF-β is a Potent Inducer of Cancer Cell Invasion in Selected Ovarian Cancer Cell Lines

Global gene expression studies by RNA-Seq analysis in anchorage-independent 3D ovarian cancer cells compared to 2D models have shown that TGF-β signaling is one of the most altered pathways in 3D spheroid models (Figure 1A) (Appendix A). Moreover, 3D spheroids can efficiently upregulate the genes involved in cell migration, Hypoxia Inducible Factor1A (HIF1A) and integrin signaling (Figure 1A,B). qRT-PCR analysis—to identify genes involved in ECM remodeling—suggests the same pattern of gene expression in 3D spheroids compared to 2D models. qRT-PCR analysis confirms an efficient rearrangement of cell adhesion genes in 3D spheroids compared to 2D models (Appendix A). It is the case that 3D spheroids overexpress a distinct group of genes in various levels, which can induce migration and invasion compared to the control group in ovarian cancer cells. These experiments established 3D cultures that can induce an invasive EMT phenotype to a limited extent. Next, the majority of the population was transformed to EMT phenotype by adding exogenous TGF-β to the 3D spheroids. The prolonged TGF-β exposure can enhance the ability of cells to maintain mesenchymal properties for a long period of time [15]. Hence, 3D ovarian cancer models were treated with a prolonged TGF-β exposure for 14 days in order to check the proliferation and efficacy of invasion capacity in the rearranged models. These 3D spheroid models treated with TGF-β have been used as model systems to study EMT and invasion.

Invasion of 3D spheroids was measured by matrigel coated Boyden Chamber Assay (Figure 1C). These cells that have undergone EMT induction by adding TGF-β and its control were used for in vitro invasion assay. Briefly, cells that migrated through matrigel and 8-micron pore size membrane towards 5% serum were then counted [16]. Cells undergone EMT have exhibited a much higher rate of migration or invasion than the control ones. Most of the ovarian cancer cell lines with epithelial phenotype show significant increase in the invasion potential of TGF-β-induced EMT condition compared to the control cells. There is a significant increase in cancer cell invasion in APOCC, SKOV3 and CAOV3 cell lines (Figure 1C). Highly aggressive teratocarcinoma cell line PA-1 and drug resistant cell line A2780 failed to show any significant difference in invasion potential compared to the other cell lines. The invasion percentage increase in all ovarian cancer cells due to TGF-β induction suggests that TGF-β is a potential inducer of migration and invasion in ovarian cancer and the EMT induction is specific to cell types.

### 2.2. TGF-β Can Induce Invasiveness and Trigger Partial EMT in the 3D Spheroids 

Seven cell lines (SKOV3, PA-1, CAOV3, A2780, A2780 CIS, APOCC, OVCAR3) were used to analyze expression pattern of the EMT signature genes in the presence of TGF-β1. The 3D spheroids treated with TGF-β for 14 days partially activate EMT by inducing some of the EMT signature genes (Figure 1D). 

From the Figure 1D, it is clear that CAOV3 is one of the most potential candidates to study cancer invasion in ovarian cancer cells. CAOV3 and SKOV3 cells significantly upregulate EMT markers like fibronectin 1 (FN1), N-cadherin (CDH2), Vimentin (VIM) and downregulate E-cadherin (CDH1) compared to the other cell lines (Figure 1D). The significant reduction in E-cadherin in CAOV3 and SKOV3 cells treated with TGF-β indicates its ability to switch cadherins (CDH1 to CDH2) and induce EMT which is essential for cancer invasion. Only CAOV3 and SKOV3 cells respond to TGF-β treatment and activate EMT partially by upregulating some EMT signature genes, such as matrix metalloproteinase (MMP2), FN1 and VIM, and downregulating E-cadherin. Each cell line responded to the TGF-β treatment in a very unique way. Invasive cancer cells exhibit a minimal alteration in EMT gene expression while epithelial-like cancer cell lines—especially CAOV3—show a higher level of expression and mesenchymal phenotype. The above results suggest that CAOV3 and SKOV3 cell lines can be potent model systems to study EMT and invasiveness in ovarian cancer cells.

### 2.3. Anchorage-Independent Models of Ovarian Cancer Cell Lines Form Distinct Type of Multicellular Spheroids 

We established anchorage-independent models for five cancer cell lines by culturing them in an ultra-low attachment plate for 14 days. Cell lines displayed different sizes of 3D structures in anchorage-independent cultures (Figure 2A). PA-1 and SW626 cell lines formed compact and large spheroids in anchorage-independent cultures. The spheroids in anchorage-independent cultures of SKOV3, OVCAR3 and CAOV3 are very loosely shaped. These cells exhibited the ability to spread and invade through the thick collagen matrix (Figure 2A). PA-1, SKOV3, CAOV3 cells showed better invasion and migration in the collagen matrix by spreading thoroughly in the thick matrix.

A2780, APOCC, CAOV3, A2780 CIS cell lines showed significant increase in the spheroid size of the TGF-β-treated multiple cell spheroids. In contrast, SKOV3 cell line displayed a significant reduction in the size of the spheroids when treated with TGF-β. Among the 3D model systems, CAOV3 and A2780 cell lines increased the spheroid size significantly compared to the other cell types (Figure 2B). The spheroids size of PA-1 and SW626 remained unaltered even after 14 days of TGF-β treatment. Obviously, TGF-β treatment increased the spheroid size and facilitated the spreading of the spheroids in the collagen matrix efficiently. PA-1 and SKOV3 cell lines showed a mesenchymal phenotype in the collagen-embedded matrix treated with TGF-β. SKOV3 and PA-1 formed star-like appearance of multicellular aggregates containing cells with spindle-like morphology which are associated with invasive phenotypes [17,18].

The size of the TGF-β-treated and control spheroids was also analyzed. The average size of the spheroids of the ovarian cancer cell lines is as following: PA-1 [433 um], SKOV3 [339 um], OVCAR3 [55 um], SW626 [149um], APOCC [50 um], A2780 [53 um], A2780 CIS [44 um], CAOV3 [26 um] (Figure 2B). Most of the cell lines displayed an increased spheroid size in the presence of TGF-β, while SKOV3 cell line showed a smaller but more compact spheroid after treatment with TGF-β. The increase in the spheroid size, with distinct zones of active growth area and necrotic center can induce more growth potential, cell viability and migration [19]. CAOV3 and A2780 cell lines have the least spheroidogenesis capacity among the cell lines that have been studied, but they were able to form spheroids of large size and better compactness after the TGF-β treatment. The compactness of spheroids was measured by checking their resistance to mechanical dissociation. Spheroids treated with TGF beta showed better resistance towards mechanical dissociation by gentle pipetting. The invasive capacity of the spheroids was evaluated by embedding the spheroids/aggregates in 3D collagen I matrices. The spheroids with more invasive ability showed better spreading and sprouting as shown in PA-1 and SKOV3 cell lines treated with TGF-β and embedded in the collagen matrix (Figure 2A). In conclusion, TGF-β treatment can increase the size and compactness of spheroids in most of the ovarian cancer cell lines studied.

### 2.4. Cell Lines Display Different Affinity Towards ECM Components

Our next aim was to study whether TGF-β treatment could alter affinity towards various essential ECM components. Not all ovarian cancer cells exhibited a similar pattern of adhesion affinity towards ECM components alone or in combination. SKOV3 cells showed the highest affinity towards ECM components as shown in the intensity plot (Figure 3A). SKOV3 is the only cell line that displayed a better attachment ability on ECM matrix in comparison to the other cell lines. SKOV3 cells showed a strong adhesion as illustrated in the intensity plot. SKOV3 cells showed a stronger affinity towards Collagen combinations, collagen1, fibronectin and laminin combinations. Other cell lines showed a limited affinity towards the ECM combinations studied. We selected the SKOV3 cell line to analyze the effect of TGF-β in altering cell adhesion affinity towards various combinations of ECM components. TGF-β facilitated adhesion of cells in different ECM components (Figure 3B). The intensity plot of cell adhesion pattern in SKOV3 cells treated with TGF-β demonstrated that TGF-β does not only facilitate adhesion towards most of the 36 ECM combinations but also facilitates migration by increasing the adhesion capacity on the promigratory ECM components like fibronectin and laminin (Figure 3B). The representative raw data image of SKOV3 cells in ECM array (Appendix A) shows that cells display a better attachment in almost all combinations treated with TGF-β with high affinity towards fibronectin and combinations of fibronectin with other ECM components. LY364947 facilitates a reversion of ECM affinity in TGF-β-treated cells.

### 2.5. TGF-β Treatment Induces Significant Cell Growth Only in 3D Spheroids of CAOV3 Cell Line

TGF-β increased cell growth percentage in CAOV3, APOCC, SKOV3, OVCAR3 and A2780 cis cell lines in 2D models compared to the other cell lines (Figure 4A). Only CAOV3 cell line showed a significant cell growth increase in both 2D and 3D models. Anchorage-independent model of CAOV3 showed an increase in cell growth under the influence of TGF-β (37%). The CAOV3 cells changed their epithelial phenotype to an elongated mesenchymal phenotype after treatment with TGF-β (Figure 4B). Although TGF-β increased invasion potential and cell adhesion pattern in different cell lines, it failed to induce cell proliferation in all cell lines except in 3D CAOV3 model. CAOV3 is the only reported cell line that had an increased growth potential as a response to the prolonged TGF-β treatment.

### 2.6. TGF-β Receptor Kinase I Inhibitor (LY364947) Delays Wound Healing and Decreases Invasion Ability in CAOV3 Cell Line 

CAOV3 and SKOV3 are the promising cell lines to study invasion and migration among the selected cancer cells both in 2D and 3D in vitro models. To validate the effect of TGF-β in these cancer cells, we used conventional 2D cultures for TGF-β inhibitor-based study. Both cell lines responded well with the TGF-β treatment and increased the invasive and migratory potential. TGF-β treatment fastened the wound healing process and facilitated invasion (Figure 4C,D,E). In contrast, the TGF-β inhibitor treatment reverted the effect of TGF-β (Figure 4C,D,E). Our study confirms that CAOV3 and SKOV3 cell lines respond well to the exogenous TGF-β and its impact can be reverted or minimized by the addition of TGF-β specific inhibitor. The results suggest that CAOV3 and SKOV3 cells are particularly sensitive to the exogenous TGF-β which can induce invasion and migration.

### 2.7. LY364947 Decreases Cell Proliferation and ATP Production in a Concentration-Dependent Manner in 2D Model

LY364947 was able to induce cell proliferation reduction in a dose-dependent manner with an IC50 value of 52.73nM in CAOV3 and 118.5nM in SKOV3 in 2D models (Figure 5A,B left panel). The treatment with LY364947 also reduced the ATP production (Figure 5A,B right panel). Since the compound can induce cell death at a very high concentration, we decided to use a lower concentration of 10uM for further study. The Oxygen Consumption Rate (OCR), basal respiration and OCR-linked ATP production were higher in SKOV3 cells treated with TGF-β compared to control and TGF-β inhibitor-treated cells (Figure 5C). The TGF-β specific inhibitor clearly reduces cell proliferation and ATP production, inhibits wound healing and decreases OCR level suggesting that TGF-β inhibitor reverses TGF-β effect in SKOV3 and CAOV3 cells.

In conclusion, we demonstrated that both 2D and 3D phenotypes are efficient models in cancer invasion and migration studies. TGF-β—a well-known cytokine for its protumorigenic properties in cancer—can induce partial EMT in SKOV3 and CAOV3 cells both in 2D and 3D models. Our results suggest that TGF-β can induce partial EMT, cell survival, increased metabolism, ECM remodeling and sensitivity to LY364947 (Figure 6). Furthermore, our results highlight the importance of TGF-β in tumor invasion in SKOV3 and CAOV3 cells and it can be proposed as a model system to study cancer cell migration and invasion.

## 3. Discussion

Results of the current study revealed that 3D anchorage-independent spheroids turned out to be an aggressive phenotype as it activates most of the EMT inducing genes and downregulates E-cadherin in the cell lines that were studied. CAOV3 and SKOV3 3D models retain a better phenotype by upregulating most of the cell adhesion molecules like collagen 1, 3 and 5 (Appendix A). As reported earlier, all the cell lines show a typical EMT signaling called cadherin switching. E-cadherin shows a downregulation, and N-cadherin upregulates in cells that undo EMT [20,21]. E-cadherin shows a downregulation, and N-cadherin shows a remarkable upregulation in all the 3D spheroid models. EPCAM—a cell adhesion signature molecule that can help in inducing metastasis by abrogating the E-cadherin activity—is also upregulated in these models. EPCAM abrogates E-cadherin activity by reducing cell-to-cell adhesion [22]. EPCAM upregulation in all the 3D spheroids shows invasive potential. Only CAOV3 cell line shows a remarkable upregulation of the SNAIL and TWIST transcription factors among all the cell lines that have been experimented (24-fold and 10.5-fold respectively). Classical cadherins are the transmembrane component of the adherens junction. Cadherins mediate cell-to-cell adhesion through their extracellular domains and connect to the actin cytoskeleton by associating with catenin through their cytosolic domain. E-cadherin is usually associated with epithelial cells. Absence of E-cadherin and activation of N-cadherin have been shown to promote motility, mesenchymal characteristics and cell invasion [20,21].

The 3D models show abnormal ECM that may promote invasion. An increase in collagen deposition or ECM stiffness alone or in combination upregulates integrin signaling and promotes survival and proliferation [23,24]. Studies show that cancer cells migrate rapidly in areas rich with collagen fibers [25]. Collagen type IV has a potent stimulatory action on angiogenesis. Dense fibrillar collagen is a potent inducer of invadopodia via specific signaling network [26]. Collagen shows a remarkable upregulation in all the cancer cell lines under study. Collagens like COL1, COL3, COL4, COL15 and COL 6 show a relative increase in the expression of 3D spheroids. It has been reported that COLI, II, V, IX have an increased deposition during tumor formation [27,28]. Increased collagen deposition can increase matrix stiffness and thereby increase ECM stiffness and eventually induce adhesion and migration [27]. In general, collagen remodels the ECM and facilitates invasion. The 3D spheroids can be good model to study collagen remodeling in cancer. Among the cell lines, SKOV3 shows a distinct pattern in cell adhesion gene expression compared to the other cell lines, where it expresses most of the collagen types (Appendix A).

RNA-Seq analysis, showing differential expression in 3D and 2D phenotypes, revealed that 3D spheroids show better cell-to-cell interaction and cell adhesion (Appendix A). Surprisingly, it upregulates the TGF-β and HIF1 pathways leading to a conclusion that metastatic potential of a cell phenotype does not depend on the proliferation rate in 3D models. Erwin et al. reported that a direct comparison of transcriptional profiling of 3D cultures and xenografts to monolayer cultures yielded upregulation of networks involved in hypoxia, TGF-β and Wnt signaling as well as regulation of EMT in colorectal cancers [29]. A recent study shows that reversal of anchorage-independent 3D spheroid into a monolayer 2D model gives a totally different phenotype from the original 2D and mimics a metastatic model [5]. A recent study suggests that ovarian spheroids show characteristics of cancer stem cells (CSCs), including expression of CSC markers, differentiation capability and tumorigenicity [30].

Our results suggest that 3D spheroids have more heterogeneity and activate tumor-promoting pathways compared to 2D models. The most noted one among those pathways is TGF-β pathway for EMT. EMT is thought to be activated in cancer cells, linked to their dissociation from the primary tumor and intravasation into blood vessels [31]. However, the impact of the EMT in cancer progression and patient survival is still far from fully understood, from skepticism [32,33,34] to fundamental support [35,36]. Some observations in mouse models suggest notes of caution on the true role of EMT in cancer progression [37,38]. Recently, Weinberg and his team experimentally verified the contribution of EMT in metastasis with direct genetic evidence of the critical role EMT processes in breast and pancreatic tumor metastasis, challenging the Fisher et al. observations about EMT [37,39,40]. The cells in the 3D structures are expressing more EMT related genes compared to 2D structures. A recent study in 3D in vitro model of mammary epithelial cells shows transition between epithelial and mesenchymal states during EMT induction by TGF-β and its reversion by withdrawing TGF-β in culture [17]. 

TGF-β is a pleiotropic cytokine that controls proliferation, differentiation, embryonic development, angiogenesis, wound healing and other functions in many cell types. TGF-β acts as a tumor suppressor in normal epithelial cells and in early stage of tumor progression. In advanced cancers, the corrupted TGF-β pathway induces many activities that lead to growth, invasion and metastasis of cancer cells [41,42,43]. TGF-β also induces tissue fibrosis through induction of various extracellular matrix proteins. Recently, numerous studies have revealed that TGF-β stimulates EMT in certain epithelial cells [42]. TGF-β induces progression of cancer through EMT which is an important step in the invasion and metastasis of cancer [44].

In our study, TGF-β seems to induce a partial EMT—like most tumor cells—in the cell lines studied in 3D spheroids. Our results suggest that TGF-β can induce partial EMT and enhance invasion potential in ovarian cancer cells. As per earlier studies depending on signaling context, epithelial cells that lose some epithelial characteristics or attain both epithelial and mesenchymal characteristics are called partial EMT [36]. The proteins of the MMP family are involved in the breakdown of ECM, promote degradation of ECM and cleave cell adhesion molecules such as cadherin, thus potentiating cell motility [45]. The real-time analysis shows an upregulation of MMPs which means that most of the cell lines we experimented induce EMT except CAOV3 and SKOV3. CAOV3 is the only cell line showing an increase in the cell proliferation rate calculated using MTS assay. It has been described that TGF-β 1 is not only responsible for EMT induction but also capable to arrest mammary epithelial cell cycle [46]. For cells to acquire mesenchymal-like phenotype, growth arrest is necessary in most cases [47]. Even though cells show less growth potential in the presence of TGF-β, they increase the invasion ability of cells in a significant manner in all the ovarian cancer cell lines that we studied. Matrigel invasion assay shows that cells are capable of migrating efficiently in the presence of TGF-β compared to control. None of the cell lines exhibit significant cytotoxicity over LY94647 at a very low concentration. Only CAOV3 and SKOV3 cells show cell growth reduction at a higher concentration of the inhibitor. Wound healing and matrigel invasion assays in CAOV3 cells exhibit more invasion arrest and less migration after treatment with TGF-β 1 at a lower concentration. Collagen embedded invasive assay shows that TGF-β treatment can enhance invasion in all the cell lines irrespective of its growth potential. TGF-β treatment significantly increases the spheroid diameter in most of the cell lines studied.

We used ECM array to investigate the interaction between TGF-β induction in cancer cells and ECM stimuli. ECM array has 36 combinations of ECM protein conditions that are deposited onto the hydrogel surface as printed array spots. Among the cell lines studied, SKOV3 shows a better adhesion capacity and responds well with TGF-β treatment. SKOV3 cells show a better and stronger attachment to many of the ECM combinations compared to the other cell lines. The intensity plot shows that SKOV3 has a better attachment compared to the other cell lines. EMT in the SKOV3 cell line is ECM component-dependent. fibronectin and laminin combination show more attachment in SKOV3 cells after TGF-β induction. Combinations of ECM show a better attachment ability compared to individual ECM components. SKOV3 cells in 3D phenotypes show EMT induction and high level of attachment towards collagens and other ECM components, which may increase the complexity in EMT and matrix stiffness. A higher level of matrix stiffness condition has been implicated in TGF-β induced EMT and stronger ECM adhesion capability of spheroids [48]. The reversion of SKOV3 cells attachment to ECM components caused by LY364947 treatment indicates that TGF-β can induce ECM specific remodeling. TGF-β treatment shows increased OCR, ATP-linked OCR and basal respiration in SKOV3 cells, and these effects were reverted while LY364947 treatment (Figure 5C). Taken together, treatment with TGF-β can induce EMT, ECM remodeling, increase metabolism, cell survival in SKOV3 and CAOV3 cells (Figure 6).

Among the cell lines studied, CAOV3 and SKOV3 are the two promising cell lines to be used as a model system for studying EMT and invasiveness in ovarian cancer cells. Both 3D and 2D models of SKOV3 and CAOV3 cells responding to TGF-β and its receptor specific inhibitor treatments signify that both cell lines are promising to study EMT or invasion in cancer. Prolonged TGF-β induction in 3D models is more promising, as it can mimic the whole tumor microenvironment and explain many cell adhesion properties. As TGF-β inhibitors are mainly targeting the tumor microenvironments with a minimal cytotoxicity, they should be used in combination with cytotoxic agents to kill the cancer cells and revert tumor favorable microenvironments. LY364947 has a potential to be a future TGF-β-targeted anti-cancer therapy, since it efficiently reverses the effect of TGF-β in cancer cells.

## 4. Materials and Methods

### 4.1. Cell Lines and Treatment

Eight ovarian cancer cell lines—PA-1, SKOV3, SW626, CAOV3, OVCAR3 (cells were obtained from American Type Cell Collection, Manassas, VA), APOCC (cell line derived from ascites fluid), A2780 and A2780 CIS (A2780 cisplatin-resistant cell line) (Sigma, St. Louis, MO, USA)—have been used for the current study. Cells were cultured in DMEM (Life technologies, NY, USA) supplemented with 10% fetal bovine serum (FBS) (Life technologies, Carlsband, CA, USA). Cell cultures were incubated at 37 °C in a humidified atmosphere, 5% CO2-injected incubator. Cell confluency and morphology were routinely checked. Cells were washed with phosphate-buffered saline (PBS) (Life technologies, USA) then fresh growth media 10% FBS DMEM- was added every two to three days. Cellular viability was determined by Trypan blue exclusion in TC20 Automated Cell Counter (Bio-Rad, Hercules, CA, USA).

### 4.2. Anchorage-Independent Growth of Multicellular Ovarian Cancer Spheroids (3D Spheroids)

Multicellular cancer spheroids (MCS) were generated from a heterogeneous parental population by seeding cells growing during log-phase in 100mm ultra-low attachment culture dishes (Corning Inc. New York, NY, USA), culturing them in DMEM /F12 supplemented with 5% FBS then incubated at 37 °C 5% CO_2_ incubator. Cells were grown in ultra-low attachment dishes for 2 weeks to make 3D anchorage-independent models. The 3D anchorage-independent models were imaged by Axio Imager 2 Research Microscope (Zeiss, Jena, Germany) and the average size of the spheroids was measured using the Zeiss ZEN Microscope Software for Microscopic Components.

### 4.3. Real-Time PCR Analysis

The relative expression of 11 genes involved in cell adhesion and EMT was determined by real-time PCR. Total RNA was isolated from cells using RNeasy Mini Kit (Qiagen, Germany). cDNA synthesis was done using first strand cDNA synthesis kit for RT-PCR (AMV) (Roche Applied Science). FastStart Universal *SYBR Green* Master (Rox) (Roche Applied Science, Indianapolis, IN, USA) was used to amplify the specific gene using cDNA primers obtained from PrimerBank (http://pga.mgh.harvard.edu/primerbank/) (Appendix A). Each real-time assay was done in triplicate and run in StepOnePlus Real-Time PCR System (Applied Biosystems, Waltham, MA, USA). The internal control gene beta-actin and the target genes were amplified with equal efficiencies. Gene expression was analyzed using relative quantification (RQ) method. The RQ method estimates the differences in gene expression against a calibrator, control without treatment.

### 4.4. ECM Cell Adhesion Assay

ECM-cell adhesion pattern of different ovarian cancer cells was analyzed by using MicroMatrix ECM Array (Microstem, San Diego, CA, USA). The array consisted of 36 different combinations of selected cell adhesion molecules. Cells were attached to the ECM combinations according to its specificity towards different ECM components. Images were extracted using a GE Typhoon Array Scanner after fixing and staining the array with TO-PRO^®^-3 Stain (life technologies). Cell adhesion patterns of TGF-β-treated and untreated samples were also analyzed by extracting the intensity of cells attached to the ECM using ImageQuant TL 8.1 Software.

### 4.5. TGF-β Treatment

Anchorage-independent spheroids and 2D cells were treated with 10ng/mL TGF-β1 (PeproTech, Rocky Hill, NJ, USA) for 2 weeks. Spheroid size was measured by Axio Imager 2 Research Microscope (Zeiss, Jena, Germany) and the average size of the spheroids was measured using the Zeiss ZEN Microscope Software. 

### 4.6. Collagen Invasion Assay

Two weeks old spheroids (both TGF-β-treated and control) were seeded on the top of collagen gel (Collagen I RatTail, Life technologies, Carlsband, CA, USA) with a concentration of 2mg/mL in chamber slides (BD technologies, San Diego, CA, USA). 

### 4.7. Cell Proliferation Assay

The effect of TGF-β on cell proliferation in different cell lines was measured by CellTiter 96^®^ AQ_ueous_ One Solution Cell Proliferation Assay (MTS) (Promega, Madison, WI, USA). Cells were grown in 2D (96 well plate BD) and 3D anchorage-independent condition using ultra-low attachment 96 well plate (Corning, Sigma-Aldrich, New York, NY, USA) with seeding density of 5000 cells per well for 3 days. The effect of cell proliferation in the presence of TGF-β and LY364947 (Tocris Bioscience, Minneapolis, MN, USA) in 2D and 3D models was measured.

### 4.8. Cell Migration-Scratch Wound Assay

The CAOV3 cells were cultured in 6-well plates with seeding density of 1 × 10^6^ cells/well. Confluent cell monolayers were disrupted by standardized wound scratching using a sterile 200-μL pipette tip and incubated in serum-free culture medium with 10ng/mL TGF-β1 in both treated and LY364947 treated cells. Migration of cells into the bare area and recovering of monolayer was evaluated 18 h, 24 h, 48 h and 72 h using Axio Imager 2 Research Microscope (Zeiss, Jena, Germany) and the average distance of the scratch at 18h was measured using Zeiss ZEN Microscope Software.

### 4.9. Cell Invasion Assay 

The transwell chamber (Corning, New York, NY, USA) was used to measure the invasive abilities of cells according to the modified protocol. Briefly, the pre-treated with or without TGF-β1 (10 ng/mL) cells were seeded (2.5 × 10^5^ cells/well) onto the top chamber of ECM-coated with 8µm pore membrane filter. ECM consisted of a combination of matrigel, collagen I and fibronectin (2:1:1) or matrigel alone (10 mg/mL). A total of 2 ml of DMEM supplemented with 10% FBS was placed in the bottom chamber as a chemoattractant. After 24 h, invasion was evaluated by counting the number of cells penetrating the membrane, counting the cells in the top chamber (noninvasive) and the bottom of the membrane (invasive). Those cells—which had undergone EMT induction either with or without the addition of TGF-β—were used for the in vitro invasion assay. Cells that migrated through the matrigel and 8-micron pore size membrane towards the medium with 5% serum were then counted [16]. The percentage of invasion was calculated by the formula [number of cells invaded/total number of cellsX100].

### 4.10. Oxygen Consumption Rate (OCR) and Extracellular Acidification Rate (ECAR) by Seahorse

The Seahorse XFe96 Analyzer (Agilent, Santa Clara, CA, USA) was used to continuously monitor OCR and ECAR. A time of 72 h prior to the experiment, 10000 cells/well were seeded in a XF96 Cell Culture Microplate in DMEM media supplemented with 2.5% FBS, 25 mM glucose, 2mM glutamine, 1 mM sodium pyruvate and cultivated at 37 °C in humidified atmosphere and 5% CO_2_. The cells were treated with TGF-β or its inhibitor with the desired concentrations, 24 h after seeding. A total of 200 mL of XF calibrator was added to each well of the XF cartridge 24 h prior to the experiment and incubated overnight at 37 °C in humidified atmosphere and 0% CO_2_. Cells were washed with PBS 30 min prior to the experiment, and 200 μL of respective XF assay medium was added per well and incubated for 45 min at 37 °C in humidified atmosphere and 0% CO_2_. For the XF Cell Mito Stress Test, XF assay medium was supplemented with 5 mM glucose and 2 mM glutamine, 1mM sodium pyruvate. After 15 min equilibration time, OCR was assessed every 10 min after 3 min mix, 3 min wait, 3 min measure—repeated 3 times after the addition of the respective compounds. The OCR was measured by XFe96 Extracellular Flux Analyzer (Agilent Technologies, Santa Clara, CA, USA) with sequential injection of 1mM oligomycin A, 1mM FCCP and 0.5 mM rotenone/antimycin A.

### 4.11. Statistical Analysis

Data represents means ± S.E.M. with at least two to three biological replicas in triplicate. Statistical analyses were determined by using Student’s *t*-test. *p* < 0.05 was considered as significant difference between groups. The half maximal inhibitory concentration (IC_50_) of LY364947 was calculated using non-linear regression method. The analysis was done using Prism Software Version 7.

## Figures and Tables

**Figure 1 ijms-20-03568-f001:**
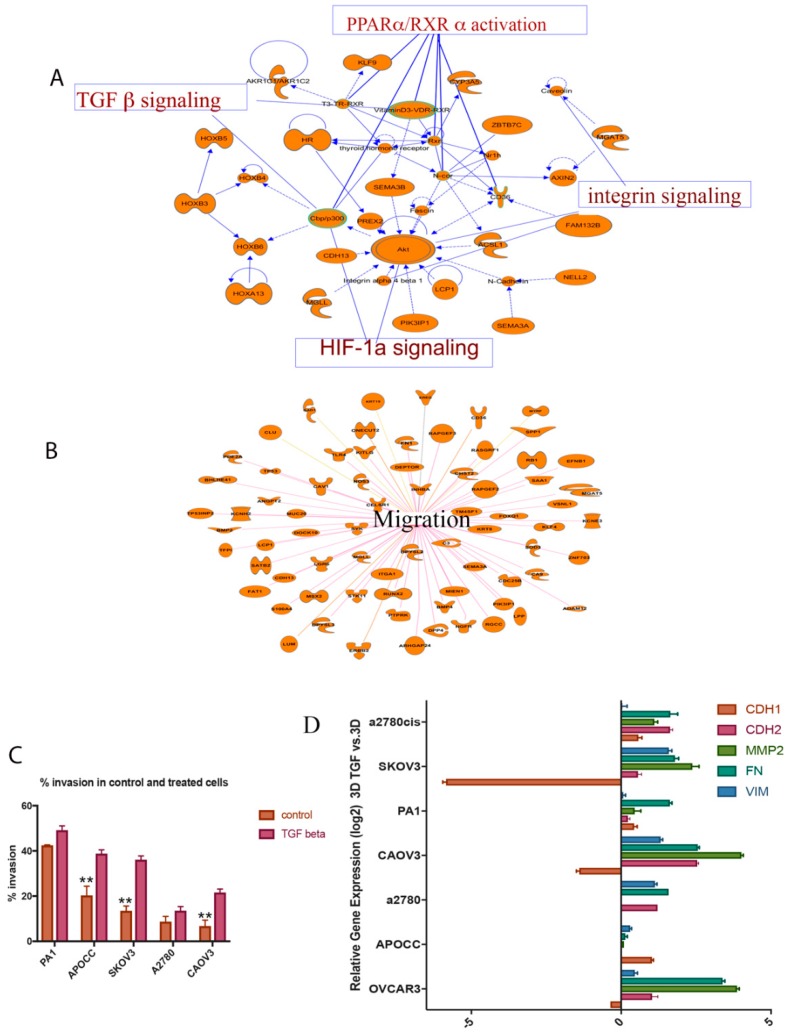
(**A**,**B**) Ingenuity® Pathway Analysis (IPA®) (QIAGEN Bioinformatics) of genes differentially expressed in 3D SKOV3 model vs. 2D and their associated pathways; (**C**) TGF-β can increase the invasion potential of the cells in a significant way. APOCC, SKOV3 and CAOV3 show a significant increase in invasion; (**D**) epithelial to mesenchymal signature gene expression pattern in 3D models treated with TGF-β compared to control 3D models. Real-time gene expression analysis shows that 3D models treated with TGF-β can activate a partial EMT in most of the ovarian cancer cell lines studied. The data in **C** is means with error bars representing Standard Error of Mean (SEM) from three experiments; ** *p* < 0.01 (Student’s *t*-test).

**Figure 2 ijms-20-03568-f002:**
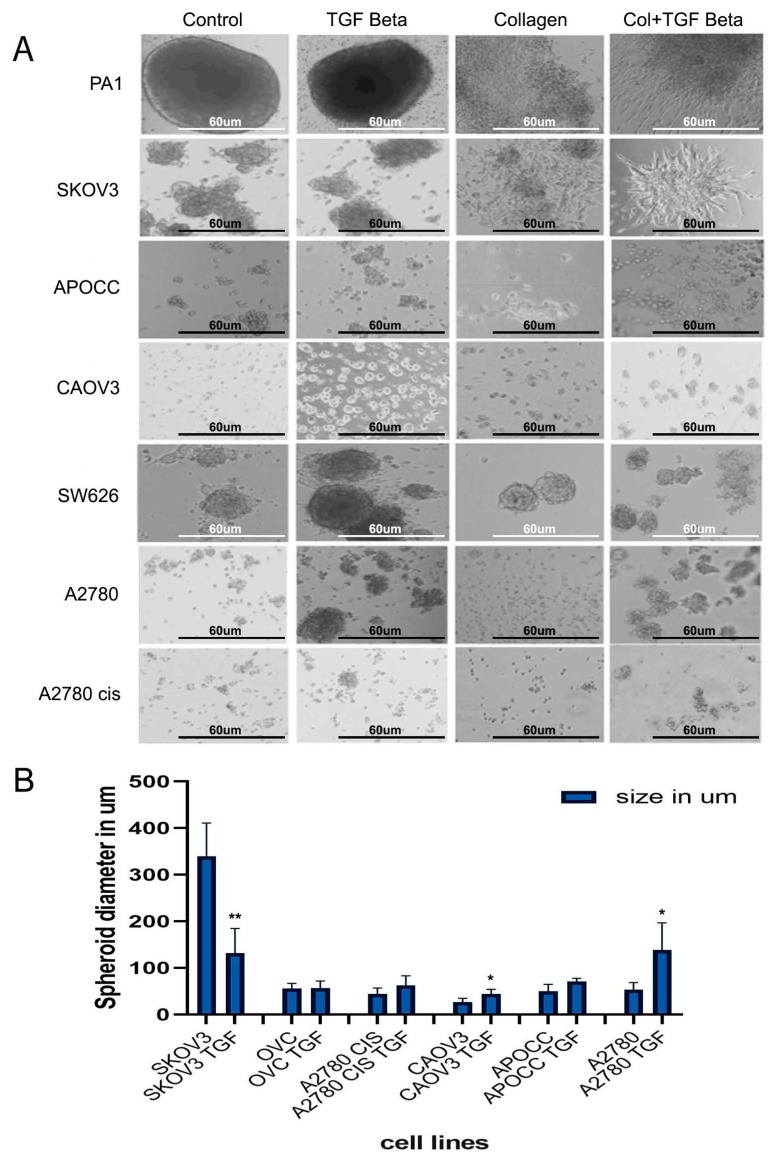
(**A**) 3D structures of various cell lines. Control and TGF-β-treated structures show different growth pattern and invasion into the collagen matrigel for most of the cell lines; scale bar = 60 um. (**B**) The size of the 3D spheroids (in diameter) before and after TGF-β treatment. The data in (**B**) is means with error bars representing Standard Error of Mean (SEM) from three experiments; ** *p* < 0.01, * *p* < 0.05 (Student’s *t*-test).

**Figure 3 ijms-20-03568-f003:**
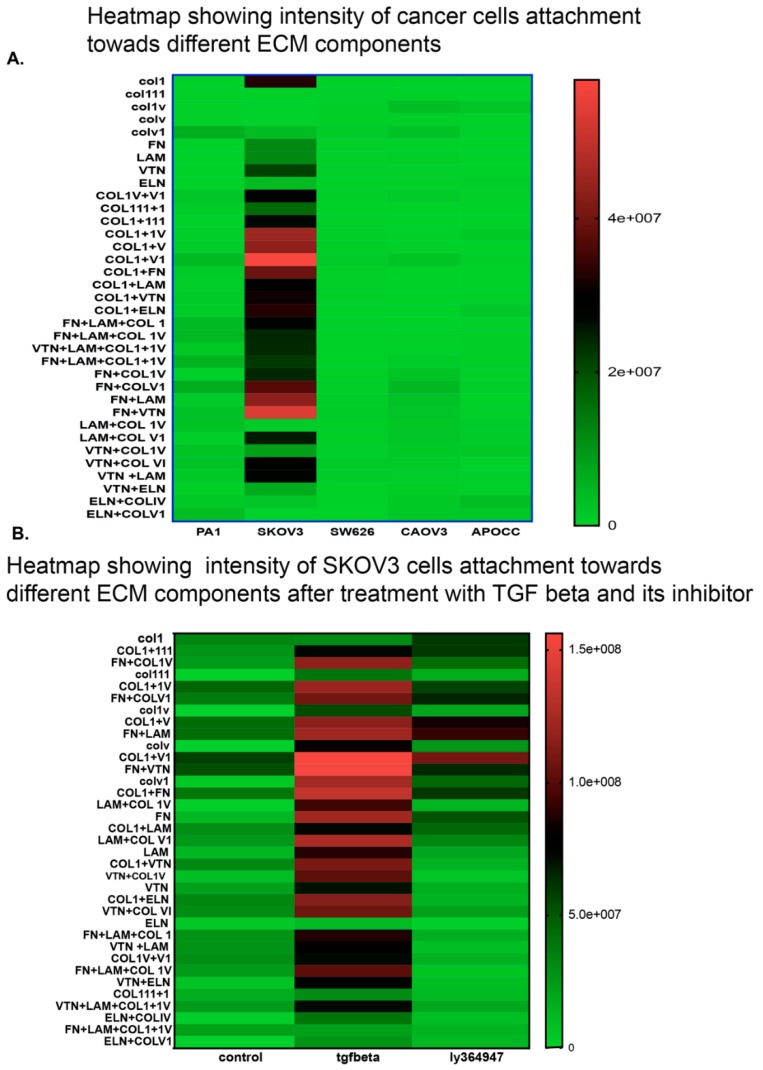
Pattern of cell adhesion on the different molecules depends on the cell type. SKOV3 shows better adhesion (**A**) and presence of TGF-β increases the cell adhesion on some ECM components more strongly in SKOV3 cells in comparison to control and TGF-β specific inhibitor LY364947 (**B**).

**Figure 4 ijms-20-03568-f004:**
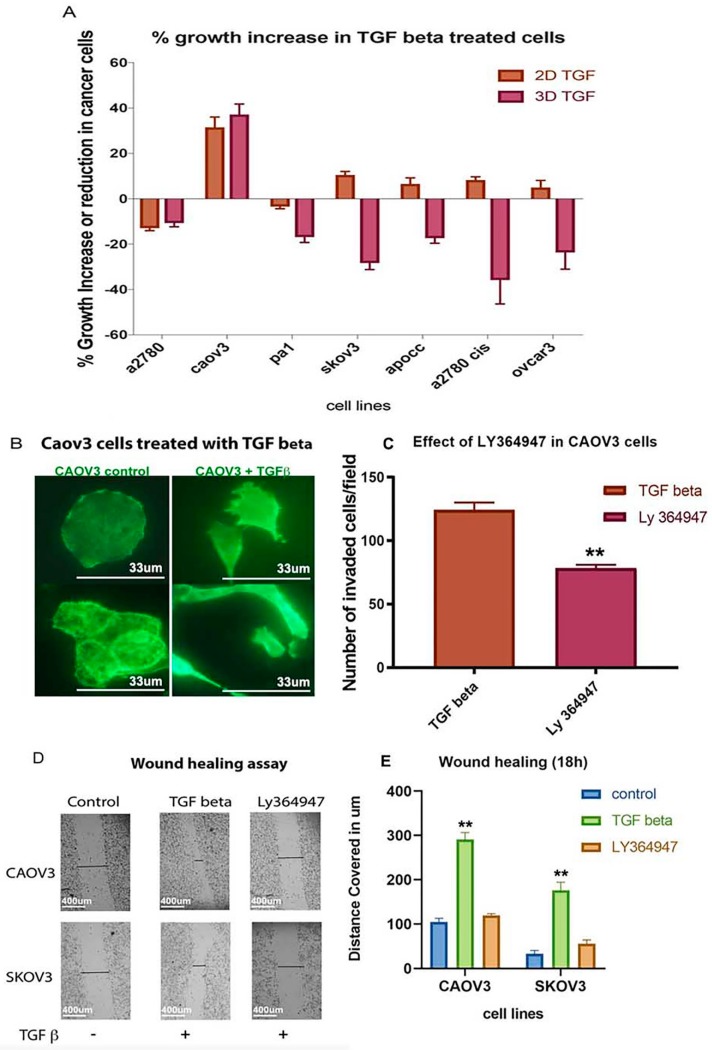
(**A**) MTS assay shows the % increase or reduction in growth potential after TGF-β treatment. Only CAOV3 cell lines show better growth progress compared to other cell lines in the presence of TGF-β. (**B**) The Epithelial to Mesenchymal Transition of CAOV3 cells after treatment with TGF-β. The image was taken by using a Zeiss inverted microscope using Actin staining. Scale bar = 33 um. (**C**) The effect of TGF-β specific inhibitor LY 364947 in cancer cell invasion. Boyden chamber assay shows reduction in the invasion potential of CAOV3 cells treated with TGF-β specific inhibitor. (**D**) A pictorial representation of the in vitro wound healing assay in CAOV3 and SKOV3 cells treated with TGF-β specific inhibitor (Ly 364947) and its control. (**E**) Graphical representation of wound healing assay in SKOV3 and CAOV3 cells. The data in (**C**,**E**) is means with error bars representing Standard Error of Mean (SEM) from three experiments; ** *p* < 0.01, * *p* < 0.05 (Student’s *t*-test).

**Figure 5 ijms-20-03568-f005:**
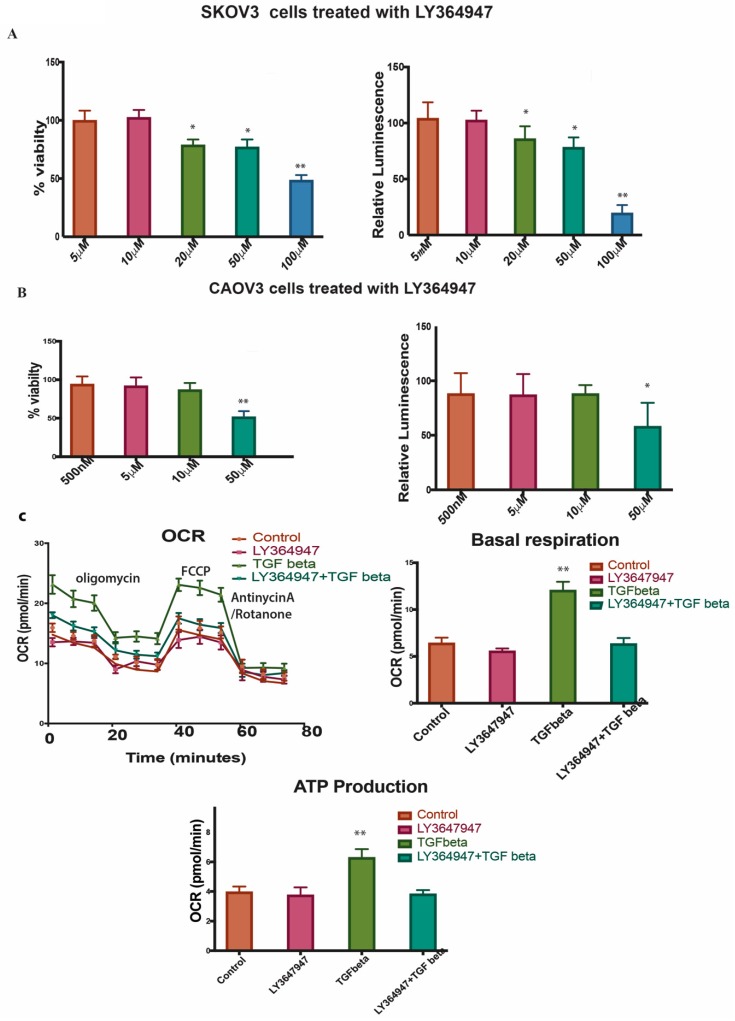
(**A**,**B**) left panel Shows % viability of SKOV3 and CAOV3 cells treated with different concentration of LY364947. The right panel shows the relative luminescence of ATP production in SKOV3 cells and CAOV3 cells treated with LY364947. (**C**) Represents seahorse assays showing oxygen consumption ratio in the top left panel, basal respiration in the top right panel, ATP production in the bottom panel for the SKOV3 cells treated with TGF-β specific inhibitor, TGF-β or in combination. The data in (**A**,**B**,**C**) is means with error bars representing standard error of mean (SEM) from three experiments ** *p* < 0.01, * *p* < 0.05 (Student’s *t*-test).

**Figure 6 ijms-20-03568-f006:**
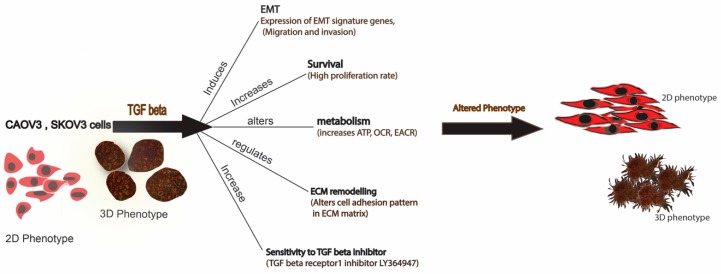
Schematic representation of the effect of TGF-β treatment in SKOV3 and CAOV3 cells. The figure represents both 2D and 3D phenotype of the cancer cells can alter its phenotype in the presence of TGF-β.

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
