# Peer review of "Cell Type-Specific TGF-β Mediated EMT in 3D and 2D Models and Its Reversal by TGF-β Receptor Kinase Inhibitor in Ovarian Cancer Cell Lines"

_ijms, 2019, doi:10.3390/ijms20143568_

Round 1

Reviewer 1 Report

The manuscript is interesting and well written. I recommend only a few minor changes:
Fig 2 A
Authors should add the scale of the bellow the photos
Fig 2B
Is it not better to give the surface of the spheres [micrometer ^ 2]?
Fig 4A
Again, I ask the authors for a scale in the drawings.
Fig 4C
I ask the authors to give the absolute value of the cell number instead of the percentage change
Fig 4D
Could the authors add a graph showing the would healing changes between each sample?

Point 4.9
The authors did not give the duration of the experiment. Please indicate after what time cells were analyzed

Author Response

The manuscript is interesting and well written. I recommend only a few minor changes:

 Thank you for the valuable comments, and suggestions. My corrections are explained below.

1.Fig 2 A
Authors should add the scale of the bellow the photos

The correction has been done in the figures.

2.Fig 2B

Is it not better to give the surface of the spheres [micrometer ^ 2]?

The spheroid diameter in um is measured using Zeiss ZEN Microscope Software.

3.Fig 4B

Again, I ask the authors for a scale in the drawings.

The correction has been done in figure 4B.

4.Fig 4C
I ask the authors to give the absolute value of the cell number instead of the percentage change

The correction has been done in the graph showing the number of cells invaded /field.

5.Fig 4D
Could the authors add a graph showing the wound healing changes between each sample?

The graph has been added (Figure 4F)

6.Point 4.9 
The authors did not give the duration of the experiment. Please indicate after what time cells were analyzed

The necessary corrections have been added in section 4.9

Reviewer 2 Report

Ameri et al. have reported in this manuscript an extensive investigation on cell-type specific TGF-β mediated Epithelial to Mesenchymal Transition (EMT) in 3D and 2D models. In addition, studies on its reversal effect by TGF-β receptor kinase inhibitor in ovarian cancer cell lines are performed. The authors have clearly identified the current limitation in tumour invasion and EMT models as well as the transcriptome profiling of 3D models. With thorough analysis and in-depth understanding, the authors have successfully investigated on the roles of TGF-β in cancer cell lines. Their studies have shown that presence of TGF-β in these cell lines caused increased invasion potential, which is specific to cell type, and trigger EMT in the 3D spheroids. With the selection of two ovarian cancer cell lines, SKOV3 and CAOV3, the authors carried out various important and well-planned studies, such as ECM cell adhesion assay, collagen invasion assay, cell proliferation assay, cell migration-scratch wound assay and cell invasion assay. It was determined that TGF-β is able to 1) induce EMT and migration, 2) increase aggressiveness, 3) increase cell survival, 4) alter cell characteristics, 5) remodel the Extracellular Matrix (ECM) and 6) increase cell metabolism favorable for tumor invasion and metastasis. To my impression, the manuscript is presented in a well-organized and logical manner. All the experimental results obtained from their studies show reasonable consistency. The results obtained from these studies have provided significant impact on the understanding of TGF-β in cancer cell lines and provide knowledge for future development of anti-cancer therapy. I would therefore recommend this manuscript for publication in International Journal of Molecular Sciences.

Author Response

Thank you for the positive and valuable comments, we will try our best to continue working on the same aspect in the future to identify more potential EMT inducing conditions in cancer cells.

Reviewer 3 Report

This is a well designed study looking into TGF-beta and EMT in ovarian cancer cell lines.

1.       The TGF-beta treatment for 14 days is not clearly described in methods, how are those 3D/2D cultures treated with TGF-beta and at what dosage. This may be relevant to different responses of different cell lines

2.       As 3D induces more EMT and invasive  charactors than 2D, this worth of discussion in revealing potential reason.

3.       The most responsive cell line SKOV3 shows stronger adhesion to collagens, which particular adhesion molecule is upregulated and responsible for this. This also raised an interesting question about the stiffness of the cancer spheres and their invasive property, needs to be discussed.

Author Response

Comments and Suggestions for Authors

This is a well-designed study looking into TGF-beta and EMT in ovarian cancer cell lines.

Thank you for the valuable suggestions. We have included major corrections and possible explanations in the revised manuscript.  

1.        The TGF-beta treatment for 14 days is not clearly described in methods, how are those 3D/2D cultures treated with TGF-beta and at what dosage. This may be relevant to different responses of different cell lines

Anchorage-independent spheroids and 2D cells were treated with 10ng/ml TGF-β1 (PeproTech, NJ, USA) for 2 weeks. It has been shown that prolonged treatment with TGF beta can induce a stable EMT condition compared to a few hours’ treatments, CAOV3 cells which has epithelial characteristics needs more prolonged TGF beta treatment to induce EMT. Considering these facts we decided to choose a longer treatment for all the cell lines studied.

2.       As 3D induces more EMT and invasive characters than 2D, this worth of discussion in revealing the potential reason.

3D spheroids can mimic the in vivo tumor condition and better model to study cancer cell invasion. Our studies revealed adding exogenous TGF beta for a prolonged period can have more EMT induction than 2D phenotypes. Since TGF beta is a pleiotropic cytokine, treatment with exogenous TGF beta in 2D phenotypes for a prolonged period may induce a hostile microenvironment.

3. The most responsive cell line SKOV3 shows stronger adhesion to collagens, which particular adhesion molecule is upregulated and responsible for this. This also raised an interesting question about the stiffness of the cancer spheres and their invasive property, needs to be discussed

 Matrix stiffness is a very important aspect in EMT, thanks for pointing out the same in our manuscript. The details of matrix stiffness and ECM attachment efficient were discussed in the discussion session of the manuscript.